# Urinary Markers of Oxidative Stress in Children with Autism Spectrum Disorder (ASD)

**DOI:** 10.3390/antiox8060187

**Published:** 2019-06-20

**Authors:** Joško Osredkar, David Gosar, Jerneja Maček, Kristina Kumer, Teja Fabjan, Petra Finderle, Saša Šterpin, Mojca Zupan, Maja Jekovec Vrhovšek

**Affiliations:** 1Institute of Clinical Chemistry and Biochemistry, University Medical Centre Ljubljana, Zaloška c.002, 1000 Ljubljana, Slovenia; josko.osredkar@kclj.si (J.O.); kristina.suhadolc@gmail.com (K.K.); teja.fabjan85@gmail.com (T.F.); petra.finderle@kclj.si (P.F.); sasa.sterpin@gmail.com (S.Š.); 2Faculty of Pharmacy, University of Ljubljana, Aškerčeva 7, 1000 Ljubljana, Slovenia; 3Department of Child, Adolescent and Developmental Neurology, University Medical Centre Ljubljana, Zaloška c.002, 1000 Ljubljana, Slovenia; davidgosar@yahoo.com; 4Center for Autism, Unit of Child Psychiatry, University Children’s Hospital, University Medical Centre Ljubljana, Zaloška c.002, 1000 Ljubljana, Slovenia; jerneja.macek@kclj.si (J.M.); maja.jekovec@guest.arnes.si (M.J.V.); 5Blood Transfusion Centre of Slovenia, Šlajmerjeva ulica 6, 1000 Ljubljana, Slovenia; mojca.zupan@ztm.si

**Keywords:** autism spectrum disorder, oxidative stress, urine biomarkers

## Abstract

*Background*: Autism spectrum disorder (ASD) is a developmental disorder characterized by deficits in social interaction, restricted interest and repetitive behavior. Oxidative stress in response to environmental exposure plays a role in virtually every human disease and represents a significant avenue of research into the etiology of ASD. The aim of this study was to explore the diagnostic utility of four urinary biomarkers of oxidative stress. *Methods:* One hundred and thirty-nine (139) children and adolescents with ASD (89% male, average age = 10.0 years, age range = 2.1 to 18.1 years) and 47 healthy children and adolescents (49% male, average age 9.2, age range = 2.5 to 20.8 years) were recruited for this study. Their urinary 8-OH-dG, 8-isoprostane, dityrosine and hexanoil-lisine were determined by using the ELISA method. Urinary creatinine was determined with the kinetic Jaffee reaction and was used to normalize all biochemical measurements. Non-parametric tests and support vector machines (SVM) with three different kernel functions (linear, radial, polynomial) were used to explore and optimize the multivariate prediction of an ASD diagnosis based on the collected biochemical measurements. The SVM models were first trained using data from a random subset of children and adolescents from the ASD group (*n* = 70, 90% male, average age = 9.7 years, age range = 2.1 to 17.8 years) and the control group (*n* = 24, 45.8% male, average age = 9.4 years, age range = 2.5 to 20.8 years) using bootstrapping, with additional synthetic minority over-sampling (SMOTE), which was utilized because of unbalanced data. The computed SVM models were then validated using the remaining data from children and adolescents from the ASD (*n* = 69, 88% male, average age = 10.2 years, age range = 4.3 to 18.1 years) and the control group (*n* = 23, 52.2% male, average age = 8.9 years, age range = 2.6 to 16.7 years). *Results*: Using a non-parametric test, we found a trend showing that the urinary 8-OH-dG concentration was lower in children with ASD compared to the control group (unadjusted *p* = 0.085). When all four biochemical measurements were combined using SVMs with a radial kernel function, we could predict an ASD diagnosis with a balanced accuracy of 73.4%, thereby accounting for an estimated 20.8% of variance (*p* < 0.001). The predictive accuracy expressed as the area under the curve (AUC) was solid (95% CI = 0.691–0.908). Using the validation data, we achieved significantly lower rates of classification accuracy as expressed by the balanced accuracy (60.1%), the AUC (95% CI = 0.502–0.781) and the percentage of explained variance (*R*^2^ = 3.8%). Although the radial SVMs showed less predictive power using the validation data, they do, together with ratings of standardized SVM variable importance, provide some indication that urinary levels of 8-OH-dG and 8-isoprostane are predictive of an ASD diagnosis. *Conclusions*: Our results indicate that the examined urinary biomarkers in combination may differentiate children with ASD from healthy peers to a significant extent. However, the etiological importance of these findings is difficult to assesses, due to the high-dimensional nature of SVMs and a radial kernel function. Nonetheless, our results show that machine learning methods may provide significant insight into ASD and other disorders that could be related to oxidative stress.

## 1. Introduction

Autism spectrum disorder (ASD) is a developmental disorder that affects social communication and behavior. People with ASD have (1) difficulty communicating and interacting with other people, (2) restricted interests and repetitive behaviors, and (3) their symptoms impact their ability to function properly in school, work, and other life domains. ASD occurs in all ethnic, racial, and socioeconomic groups and represents in the vast majority of cases a lifelong disorder. In the Diagnostic and Statistical Manual of Mental Disorders (DSM-IV) and the International Classification of Mental and Behavioral Disorders (ICD-10), autism was part of the Pervasive Developmental Disorders diagnostic category (Pervasive Developmental Disorders, PDD) and was characterized by a severe and pervasive impairment in social interaction, behavior, and communication [1]. The Diagnostic and Statistical Manual of Mental Disorders, 5th Revision (DSM-5) defines autism spectrum disorder (ASD) as a neurodevelopmental disorder with a spectrum of qualitative impairments in social interaction, qualitative impairments in communication, and restricted and stereotyped patterns of behavior, interests, and activities [2]. A diagnosis of ASD is given based solely on clinical features. At present, there are consistently validated biomarkers for diagnostic and/or screening purposes [3].

Increased vulnerability to oxidative stress by endogenous or environmental pro-oxidants in conjunction with genetic susceptibility factors may lead to the development and clinical manifestations of autism.

The shift in the balance between oxidants and antioxidants in favor of the former is termed oxidative stress (OS) [4].

OS is a pathological condition which is due to an abnormal increase of reactive species (RS), called free radicals, and a concomitant reduction of antioxidant defenses against free radicals. It is widely accepted that an excess of RS is toxic and damages essential cell components such as nucleic acids, proteins and lipids, leading to cell death. RS can lead to the oxidation of amino acid side chains, the formation of protein–protein cross-linkages, and the oxidation of the protein backbone resulting in protein fragmentation. The body produces a defined amount of RS due to the physiological cell metabolism. RS are dangerous because they have the spontaneous tendency to oxidize molecules due to their capacity to capture an electron or a hydrogen atom from any substance that gets in contact with them. An imbalance in the oxidant/antioxidant relationship such as an overproduction of RS can become toxic to neurons by inducing DNA methylation [5] and damage to various tissues [6]. One of the most common mechanisms by which RS attack molecules of our body is the generation of so-called “hydroperoxides” or ROOH. RS are neutralized by anti-oxidants, which are used by the body to protect it from the damaging effects of free radicals. Anti-oxidant enzymes represent a wide range of enzymes that transform RS into harmless substances. Oxidative damage produced by reactive oxygen species (ROS) has been implicated in the etiology and pathology of many health conditions, including a large number of chronic diseases.

A number of studies have reported evidence of OS in individuals with ASD [7,8,9,10,11,12,13,14], implicating OS a contributory factor in the development of ASD. Its impact on individuals with ASD may be caused by a range of RS. RS could come from external sources (e.g., environmental pollution, metabolism of xenobiotics) or could be generated in the body by NADPH oxidases [12,13,15,16]. RS includes superoxide (O_2_^●−^), hydroxyl, peroxyl, alcoxy, and nitric oxide (NO) free radicals.

Biomarkers of OS are measured in the blood/plasma, urine, and saliva, with urinary biomarkers representing the least invasive way to assess OS in the individual patient. Urinary biomarkers of oxidative status present a great opportunity to study redox balance in human populations. In addition, specimen collection is non-invasive, and urine offers a better matrix than blood/plasma since it contains a lower organic and inorganic metal content. Also, urinary levels of the biomarkers present intergraded indices of redox balance over a longer period of time compared to blood levels.

The most common oxidative markers of protein oxidation are protein carbonyls and protein nitrates. The common markers of lipid peroxidation by free radicals are malonyldialdehyde (MDA), 4-hydroxynonenal (HNE), and F2-isoprostanes. The nucleic acid oxidation products that are frequently used for measurement of oxidative stress are 8-hydroxy deoxyguanosine for DNA, and 8-hydroxyguanosine for RNA.

Although extensive studies have been carried out on autism, there are still controversies regarding its underlying mechanisms. Evidence of neurodegeneration has been seen in some cases of children with ASD who experienced a progressive loss of neurological function, with signs of elevated 8-oxoguanine levels, which could be evidence of oxidative stress [17]. In our study, we focused on four relevant biochemical markers of OS.

### 1.1. 8-Hydroxy-2′-Deoxyguanosine (8-OH-dG)

Lipids of cellular membranes, proteins, and DNA are regularly damaged by ROS. While studying their impact on cellular biomolecules, major efforts have been devoted to understanding how RS attack and modify DNA [18,19,20], including ROS’s induction of aberrant DNA methylation [21]. Among the most studied topics in this respect is 8-hydroxy-2’-deoxyguanosine (8-OH-dG) [20], a product of oxidatively damaged DNA formed by hydroxy radicals, singlet oxygen and direct photodynamic action. 8-OH-dG can be detected in tissues, serum, urine and other biomaterials. 8-OH-dG has been validated as a biomarker for oxidative stress in animal models, but so far, this has not been shown in human studies. Previous research has validated it as a sensitive biomarker of DNA damage due to oxidative stress [22,23]. The impact of inter-individual variability in DNA repair capacity on the levels of 8-OH-dG in urine remains unknown. Intra-individual variability is also uncertain. However, this biomarker can be useful as a monitoring tool.

### 1.2. F_2_-Isoprostanes

F_2_-isoprostanes are formed during the non-enzymatic oxidation of arachidonic acid by different types of free radicals [24]. F_2_-isoprostanes and their metabolites, excreted in the urine, are chemically stable compounds [24] and are not sensitive to the dietary intake of lipids [25]. Among them, 8-isoPGF2α is an especially reliable indicator of lipid peroxidation, due to its stability. The measurement of F2-isoprostanes in bodily fluids is regarded as a trustworthy biomarker for the in vivo determination of lipid peroxidation by free radical pathways [24,25,26,27]. This is a well-studied and validated biomarker of oxidative status, with known inter- and intra-individual variations [28]. However, there is some concern that inter-individual variation in F2-isoprostane hydrolysis form esterified lipids could contribute to the differences in their steady-state urinary levels.

### 1.3. Dityrosine (DT)

Tyrosine is one of the major targets of protein oxidation. DT is a tyrosine dimer, derived from tyrosyl radicals, which is formed by RS, metal-catalyzed oxidation, ultraviolet irradiation, and peroxidases. DT has previously been proposed to be a good indicator of protein oxidation [15,29]. When measured in urine, it is sensitive to pathological conditions associated with oxidative stress [30,31,32,33]. Promising understudied biomarkers, such as dityrosine, reflect oxidative protein damage. The addition of this biomarker to the panel of validated indices of oxidative status diversifies the existing tools in studying the role of oxidative status in human health and disease.

### 1.4. Hexanoyl-Lysine (HEL)

Hexanoyl-lysine (HEL) adduct is a novel lipid hydroperoxide-modified lysine residue. HEL is formed by oxidative modification by oxidized omega-6 fatty acids such as linoleic acid or arachidonic acid. HEL can be a useful biomarker for initial stages of lipid peroxidation [34]. Urinary levels of HEL were also found to be elevated in children with ASD [35], with a recent study by Yui et al. showing that increased HEL is associated with decreased antioxidant capacity in children with ASD [36].

The aim of the present study was to examine the levels of OS markers in the urine of children and adolescents with ASD in comparison to their healthy peers. We also wanted to examine if differences in OS markers could be used in combination to differentiate both groups using modern machine learning methods, in part replicating earlier efforts by Anwar et al. [37] to use support vector machines (SVMs) to predict the presence of ASD based on different biomarkers.

## 2. Subjects and Methods

### 2.1. Patients

Our group of patients was composed of 139 children and adolescents with autism spectrum disorder (ASD). The children and adolescents were diagnosed previously in a tertiary out-patient clinic by a developmental pediatrician or psychiatrist and psychologist. Children and adolescents were diagnosed with ASD based on four behavioral criteria: (1) deficits in social communication and interaction (including social-emotional reciprocity, nonverbal communication, and development/maintenance of relationships); (2) restricted, repetitive patterns of behavior, interests, or activities (such as movements like arm flapping, excessive adherence to routine or rituals, resistance to change, obsessive fixations, and sensory sensitivity); (3) presentation of symptoms during early childhood; and (4) impairment of everyday functioning [1].

The diagnosis was made using a multidisciplinary approach that combined a clinical evaluation along with a psychological assessment. Children were grouped according to the criteria detailed and summarized by DSM-5 [38]. Additional behavioral ratings were based on a standardized classification of behavior for children with ASD developed by the local educational authority for providing additional school support [39]. Ratings were given for two separate dimensions: a) presence of deficits in social communication and social interaction, and b) presence of deficits in behavioral flexibility and of limited interests and activities. Each child in the ASD group received a rating on each of the two dimensions on a three-point rating scale (1: mild deficit; 2: moderate deficit; 3: severe deficit). Children in the control groups received a rating of 0 for both dimensions.

A group of children and adolescents (47) without any neurological problems or any other known acute or chronic disease was used as a control group. Before inclusion of the children and adolescents, we obtained written permission from their parents.

The demographic characteristics of participating children and adolescents are given in Table 1. The design of our study was approved by the National Medical Ethics Committee (0120-201/2016-2 KME 78/03/16).

### 2.2. Methods

We obtained a second-morning urine specimen from all participants. Urine samples were aliquoted immediately after their collection and frozen to a temperature of −80 °C until they were analyzed. Before analysis, urine was defrosted and centrifuged.

For the determination of three of the four biochemical parameters, we used commercial ELISA lits: 8-OH-dG Check ELISA, HEL ELISA and the DT ELISA kit provided by the Japanese Institute for the Control of Aging (JaICA, Japan). 8-izoprostane was determined using a kit from IBL International (Hamburg, Germany). Urinary creatinine was obtained using the Roche reagent kit on a Roche Modular P analyzer (Roche Diagnostics GmbH, Mannheim, Germany). All tests were performed following the instructions of the kit provider.

## 3. Statistical and Machine Learning Analysis

All statistical analyses were done using the R+ language for statistical computing [40]. As part of the descriptive statistical analysis, we described the expression of our four biochemical markers normalized by urinary creatinine levels (8-OH-dG, 8-isoprostane, dityrosine, HEL) by calculating their median and inter-quartile range in our ASD in control groups, respectively. We then compared their distribution across both groups using the non-parametric two-sample Wilcoxon test.

In the second step of our analyses, we used support vector machines (SVMs) to predict an ASD diagnosis based on the expression of the four chosen biomarkers. We did this using the “caret” R+ package [41]. We examined the predictive power of three types of SVMs, with each type using either a linear, radial or polynomial (up to 3rd-degree polynomial) kernel function. We optimized the model hyper-parameters on a random subset of our data. This random subset of data represented the training data for the SVMs. It included 70 children and adolescents from the ASD group (90% male, average age = 9.7 years, age range = 2.1 to 17.8 years) and 24 from the control group (45.8% male, average age = 9.4 years, age range = 2.5 to 20.8 years). In order to tune our SVMs to the training data, we used the optimism bootstrap estimator (1000 samples per SVM) with the default grid search settings and accounted for the unequal numbers of participants across our research groups by using the synthetic minority over-sampling technique (SMOTE) [42].

We evaluated the predictive performance of our SVMs in the context of the training data by looking at the achieved balanced accuracy (combined estimate of true positives and true negatives) and the Cohen’s Kappa coefficient. In addition, we estimated the explained variance (*R*^2^) based on the Spearman rho correlation coefficient and performed significance testing for the estimated correlation coefficient [43]. To ascertain the importance of individual variables, we also looked the variable importance metric (VI), which ranges from 0 to 100 [44]. Finally, we depicted the predictive accuracy of the SVMs using ROC curves and estimating the 95% confidence interval for the area under the curve (AUC). We then repeated these analyses for SVM prediction accuracy using an independent validation sample, which was composed of 69 children and adolescents from the ASD (88% male, average age = 10.2 years, age range = 4.3 to 18.1 years) and 23 from the control group (52.2% male, average age = 8.9 years, age range = 2.6 to 16.7 years), who were not among those whose data was included in the training dataset. This provided a crucial test of the reproducibility of our results on data not previously seen by our SVMs.

Finally, to further validate our findings, we examined the correlation between the predicted probability of having an ASD diagnosis based on the SVM models and the ratings of deficits in the domains of social communication and behavioral flexibility. These correlations were estimated by calculating the Spearman correlation coefficient.

## 4. Results

Our univariate analyses indicated a somewhat higher variability of biomarker expression in the ASD group. The only trend towards a difference in the overall levels of expression between the ASD and control groups was seen for 8-OH-dG, with lower levels being more typical of children and adolescents in the ASD group. The results are presented in Table 2.

Among all of the examined SVMs, only the SVM with a radial kernel function proved to accurately predict an ASD diagnosis (see Table 3). This SVM predicted an ASD diagnosis with a balanced accuracy of 73% and explained a significant amount of variance. It also proved to be the only SVM that was able to predict an ASD diagnosis with any accuracy in the validation dataset.

Despite their relatively poor predictive power in the context of the validation data, our results offer some insight into the importance of individual biomarkers in predicting a diagnosis of ASD. While investigating the standardized variable importance of each biomarker in the SMV with a radial kernel, we found that the most important predictors were 8-OH-dG (VI = 100.00) and dityrosine (VI = 73.94), followed by 8-isoprostane (VI = 16.20), and HEL (VI = 0.00). When using predictions from the SVM with the radial kernel function to plot the likelihood of receiving an ASD diagnosis in relation to 8-OH-dG and dityrosine, we found that the control group was predicted to have an optimal level of 8-OH-dG expression at around 2.5 to 3.0 standardized units (see Figure 1). Our second-most important predictor in the radial kernel SVM showed a somewhat different trend. Higher levels of dityrosine expression tended to be associated with a higher likelihood of receiving an ASD diagnosis. Additional information on the predicted probabilities of receiving a diagnosis based on the SVM with the radial kernel function are presented in Figure A1 and Figure A2 in Appendix B.

When examining the association between the predicted probability of having an ASD diagnosis and ratings of deficits in social interaction and behavioral flexibility within the training dataset, we found several significant correlations for the SVM model with the radial kernel function. However, we could not verify these correlations within the validation sample (Table 4).

## 5. Discussion

The results of our study indicate that, among the four biomarkers used, 8-OH-dG, a biomarker of DNA damage due to oxidative stress and an individual’s DNA repair capacity [20,45], and dityrosine, an oxidation damage marker and indicator of protein oxidation [29,44], proved to be the most influential predictors of having an ASD diagnosis in the group of children and adolescents included in our study. The predictive accuracy of our SVM algorithm was not as high as that of the SVM algorithms published in a previous study by Anwar et al. [37], and was also significantly reduced in an independent validation dataset. Although far from having the potential of being a reliable diagnostic tool, our SVMs did have some predictive accuracy in the validation data set and offered some interesting insights into the role of oxidative stress in the etiology of ASD.

Our examination of the SVM predicted probabilities of having an ASD diagnosis across a range of biomarker expression, which indicates a potentially interesting finding in relation to the measurement and involvement of 8-OH-dG in ASD. The superior performance of the SVM using a radial kernel function indicates that individuals with ASD might have levels of 8-OH-dG below or above an optimal range [46,47], rather than simply levels that are higher than neurotypical individuals. This could explain some the disparate findings of previous studies, among which some have found increased levels of 8-OH-dG in individuals with ASD [12,48,49], while others have not [35,36,50]. In individuals with ASD, such deviations from an optimal level of 8-OH-dG could be a reflection of a departure from an appropriate range of ROS expression, which has been shown to be important in regulating neuronal differentiation, migration [51,52], and synaptic pruning [53,54], which are all processes thought to be involved in the etiology of ASD [55,56]. It could also be an indicator of the potential genetic vulnerability of children with ASD to oxidative stress [12,57,58,59], environmental factors contributing to it [12,60,61], as well as deficient mechanisms of DNA repair [62].

The second-most import predictor of diagnostic status in our radial kernel SVM was dityrosine. This result is in line with a previous study that used an SVM algorithm with a linear kernel function to classify children according to ASD status. The study by Anwar et al. [37] found dityrosine to be an important predictor of ASD, and as in our study, ASD diagnosis was associated with higher levels of dityrosine expression. This finding also fits well with other studies implicating tyrosine metabolism in the etiology of ASD [63,64,65].

The remaining two biomarkers used in our study, hexanol-lysine and 8-isoprostane, did not have high variable importance in our SVM that used a radial kernel function. This does not mean they have no role in the study of oxidative stress in ASD, but merely that they did not provide much additional information in the specific set context of our biomarkers and data. However, they have in other studies been shown to have aberrant levels of expression in individuals with ASD [28,35,50].

We hope our study also adds to the current literature from a methodological standpoint. We feel our use of an independent validation dataset somewhat guards against the potential overfitting of data that can occur when using SVMs, especially those using a radial kernel function. We also hope that our results illustrate the reduction in predictive accuracy when SVMs are employed on data which are not part of the dataset used for SVM model tuning. In light of the public interest in such studies, it would seem important to emphasize that fact to the general public, as pointed out by Buxbaum et al. [66].

In regard to our own study, we would also like to note some other limitations. Due to a discrepancy in the number of individuals in the ASD and control groups, we had to employ SMOTE sampling procedures [42] to control for the uneven number of participants across groups when tuning our SVM models. If we had the resources, even more data for the control group would have been preferable, making the numbers of participants in each of our research groups more comparable. The relatively small number of participants in our study also limits our ability to generalize the results across the diverse population of individuals with ASD. We therefore feel that the interpretation of our findings would greatly be enhanced by the replication of our results on new data sets by other research groups. We therefore provide the R+ code for the SVM calculations used in our study in Appendix A, hoping researchers might utilize the SVM algorithms and settings used in future studies.

We are aware that our groups are different in terms of gender representation. The parameters that we have determined in urine are not subject to gender differences, and therefore this difference does not introduce an additional bias. The concentrations and profiles of oxidative stress markers are not significantly affected by gender, age, or body mass index (BMI) [67,68,69,70,71,72].

When looking at the literature on oxidative stress in ASD from a wider methodological viewpoint, we also feel that the field would benefit from a greater effort in obtaining larger datasets, beyond those currently available. This would seem especially prudent if the field is intent on using machine learning methods to gain greater insight into the etiology of ASD. A multi-center collaborative study of oxidative stress in ASD, similar to those in the field of ASD neuroimaging [73,74], might represent a way to gather data from hundreds of participants in order to make the predictions of machine learning algorithms more reliable and informative. By including genetic and neuroimaging data, such efforts could help deepen our understanding of the aberrant neurodevelopmental processes that the biomarkers used in this study measure in individuals with ASD.

## 6. Conclusions

Support vector machines (SVMs) offer great promise for enhancing the methodological toolset used in studying oxidative stress in ASD, if used and interpreted properly [74,75]. The ability of SVMs to recognize nonlinear boundaries in the multi-dimensional space formed by measurements of biochemical markers could be an advantage when trying to identify aberrant biochemical processes in ASD. In our study, SVMs allowed us to improve our prediction of an ASD diagnosis based on the utilized biomarkers and granted us greater insight into the measurement of biomarker expression. We discovered that, in addition to identifying a pathological level of biomarker expression, it may also be important to identify potential deviations for an optimal set-point of biomarker expression using SVMs. We hope our work will encourage others to replicate the use of SVMs in this field of research, and we look forward to this line of inquiry having an impact on the practice of ASD diagnosis and treatment.

## Figures and Tables

**Figure 1 antioxidants-08-00187-f001:**
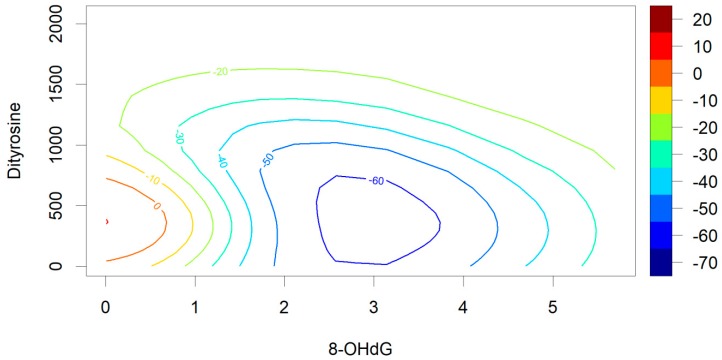
Increase or reduction in probability of an ASD diagnosis (in %) based on support vector machine (SVM) predictions from 8-hydroxy-2’-deoxyguanosine (8-OH-dG) and dityrosine. Predictions shown for an SVM using a radial kernel.

**Table 1 antioxidants-08-00187-t001:** Demographic characteristics of children in autism spectrum disorder (ASD) and control groups.

Characteristics	ASD	Control
N	139	47
Male	89%	48%
Mean age (in years)	10.0	9.1
Age range (in years)	2.1–18.1	2.5–20.8
Deficits in social communication		
Mild	47%	/
Moderate	47%	/
Severe	6%	/
Deficits in behavioral flexibility		
Mild	48%	/
Moderate	43%	/
Severe	9%	/

**Table 2 antioxidants-08-00187-t002:** Urinary markers normalized by creatinine in the ASD and control groups. IQR: interquartile range; Me: Median; W: Wilcoxon signed-rank test.

Marker	ASD Group (*n* = 139)			Control Group (*n* = 47)			*W*	*p*
	*Me*	*IQR*	*Range*	*Me*	*IQR*	*Range*		
8-OH-dG	1.12	1.42	0.05–22.81	1.57	1.24	0.01–6.38	2716.00	0.085
8-isoprostane	143.72	525.42	2.86–8000	130.94	436.90	3.48–2026.96	3241.50	0.939
Dityrosine	268.04	338.61	0–6475.71	171.94	269.16	4.05–877.83	3722.00	0.179
Hexanoil-lysine	5.84	5.35	0.5–97.39	5.44	5.14	1.62–30.21	3291.00	0.940

**Table 3 antioxidants-08-00187-t003:** Performance evaluation of support vector machines (SVMs) for predicting ASD on training and validation data.

SVM Kernel Function	Balanced Accuracy	*Kappa*	*p*	*R* ^2^	*p*	AUC	95% CI for AUC
Linear (training data)	50%	0.000	0.999	0.000	0.999	0.404	0.270–0.537
Radial (training data)	73%	0.455	0.823	0.208	0.001	0.799	0.691–0.908
Polynomial (training data)	57%	0.129	0.860	0.017	0.458	0.643	0.518–0.767
Linear (validation data)	50%	0.000	0.999	0.000	0.999	0.579	0.443–0.715
Radial (validation data)	60%	0.194	0.071	0.038	0.174	0.641	0.502–0.781
Polynomial (validation data)	55%	0.093	0.391	0.009	0.663	0.623	0.490–0.755

**Table 4 antioxidants-08-00187-t004:** Rho correlations between SVMs’ predicted likelihood of ASD diagnosis and deficits in social communication and behavioral flexibility based on training and validation data.

Symptom Domain	Linear		Radial		Polynomial	
	*rho*	*p*	*rho*	*p*	*rho*	*p*
Deficits in social communication						
Training data	−0.128	0.337	0.418	0.001	0.187	0.223
Validation data	−0.093	0.755	0.239	0.120	0.242	0.120
Deficits in behavioral flexibility						
Training data	−0.143	0.337	0.446	0.001	0.198	0.223
Validation data	−0.061	0.755	0.234	0.120	0.207	0.144

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
