# Peer review of "Urinary Markers of Oxidative Stress in Children with Autism Spectrum Disorder (ASD)"

_antioxidants, 2019, doi:10.3390/antiox8060187_

Reviewer 1 Report

The authors that are examined the Autism spectrum disorder (ASD), which is a developmental disorder characterized by deficits in social interaction, restricted interest and repetitive behavior. They are proposed that this defected is correlated to

Oxidative stress response in young people. Then, authors analyzed the possible oxidative stress biomarkers such as 8-OH-dG, 8-isoprostane, dityrosine and hexanoil-lisine from urinary by ELISA assay. They found that urinary levels of 8-OH-dG and 8-isoprostane are predictive of an ASD diagnosis. Some comments as follows:

Minor comments:

1. The Austin disease is correlated to brain damage, but why is collect the urine sample and checked some molecules such as the 8-OH-dG, 8-isoprostane, dityrosine and hexanoil-lisine that are how to correlate to this defected should be mention in Introduction.

2. If this diseases is correlated to ROS generation that may treat and supply the antioxidants such NAC or GSH in the clinical should be improved.  

3. In the test samples may consider probed some oxidative stress antioxidant enzymes such as the Cu/ZnSOD or catalase is closed to ROS response.

Author Response

Reviewer 1

1. The Austin disease is correlated to brain damage, but why is collect the urine sample and checked some molecules such as the 8-OH-dG, 8-isoprostane, dityrosine and hexanoil-lisine that are how to correlate to this defected should be mention in Introduction.

 Reviewer wishes to clarify the correlation between our biochemical parameters and neurobiological changes in autism. We made these explanations in the discussion, which can be pointed out.

2. If this diseases is correlated to ROS generation that may treat and supply the antioxidants such NAC or GSH in the clinical should be improved.  

In this point, it is suggested that treatment in the clinic with anti-oxidants may be proposed, but is neglected the baseline finding of results, that our findings are far from accurate and do not permit such detailed conclusions. Reviewer also proposes additional biochemical parameters for measurement, which in no way contributes to the replication of the Anwar study and others (2019), which we wanted to repeat.

3. In the test samples may consider probed some oxidative stress antioxidant enzymes such as the Cu/ZnSOD or catalase is closed to ROS response.

These parameters are determined in erythrocytes; we had only urine samples.

Reviewer 2 Report

Major issues:

- On biomarkers discussion, also add: Behav. Sci. 2019; 9:5.

- Provide references for lines 67-69.

- It should be clear that the outcomes of this study are indicated to be predictive, before claim the effectiveness use of these biomarkers for ASD diagnosis.

- The levels of 8-OHdG in ASD subjects, even if lower than controls, appear to be not significative. If so, discuss the utility of the analysis of this marker by a biological point of view.

Author Response

Reviewer 2

On biomarkers discussion, also add: Behav. Sci. 2019; 9:5.

Although extensive studies have been carried out on autism, there are still controversies regarding its underlying mechanisms. Evidence of neuro degeneration has been seen in some cases of children with ASD who experienced progressive loss of neurological function, with the sign of elevated 8-oxoguanine levels, what could be the evidence of oxidative stress 

[17] Mohamed B. Abou-Donia et al. de novo Blood Biomarkers in Autism: Autoantibodies against Neuronal and Glial Proteins. Behav. Sci. 2019, 9, 47; doi:10.3390/bs9050047;

Kern, J.K. Geier, D.A.  Sykes, L.K.Geier, M.R. Evidence of neurodegeneration in autism spectrum disorder. Transl. Neurodegener. 2013, 2, 17

In the sentence (specific volume of the journal) on biomarker discussion there is no strict connection to our study, so we didn’t citated them:

Steroid Metabolites Support Evidence of Autism as a Spectrum by Benedikt Andreas Gasser, Johann Kurz, Bernhard Dick and Markus Georg Mohaupt, Behav. Sci. 2019, 9(5), 52; https://doi.org/10.3390/bs9050052

Caregiver Reports of Screen Time Use of Children with Autism Spectrum Disorder: A Qualitative Study by Anja Stiller, Jan Weber, Finja Strube and Thomas Mößle, Behav. Sci. 2019, 9(5), 56; https://doi.org/10.3390/bs9050056

Provide references for lines 67-69.

We added the reference:

[1] American Psychiatric Association: Diagnostic and Statistical Manual of Mental Disorders, 4th ed. Washington: American Psychiatric Association; 1994

It should be clear that the outcomes of this study are indicated to be predictive, before claim the effectiveness use of these biomarkers for ASD diagnosis.

The reviewer has responded well to our corrections so far and is mostly satisfied and wants minor changes and quotes and warnings about limited predictive validity of our findings, which is understandable in light of the context of our study.

The levels of 8-OHdG in ASD subjects, even if lower than controls, appear to be not significative. If so, discuss the utility of the analysis of this marker by a biological point of view.

In this point, it is pointed out that the differences in 8-OHdG are not important, but we regret not taking this into account, whatwe wrote in the discussion. The explanation we gave was that the values in the group of children with autism move outside the optimum range of the control group (ie they are lower and higher), which leads to the fact that only a statistical test that checks the difference in one direction can not be detected ( either larger or smaller), while radial pinch (kernel) detect them. In this discussion, we also give a biological interpretation of the significance of such deviations (that is, the values outside of the optimum) and we base them on the literature (paragraphs 267-279)…

Our examination of the SVM predicted probabilities of having an ASD diagnosis across a range of biomarker expression indicates a potentially interesting finding in relation to the measurement and involvement of 8-OHdG in ASD. The superior performance of the SVM using a radial kernel function indicates that individuals with ASD might have levels of 8-OHdG bellow or above an optimal range [43,44], rather than just levels that are higher than neurotypical individuals. This could explain some the disparate findings of previous studies, among which some have found increased levels of 8-OHdG in individuals with ASD [11,45,46], while others have not [32,33,47]. In individuals with ASD such deviations from an optimal level of 8-OHdG could be a reflection of a departure from an appropriate range of ROS expression, which has been shown to be important in regulating neuronal differentiation, migration [48,49] and synaptic pruning [50,51], all processes thought to be involved in the etiology of ASD [52,53]. It could also be an indicator of the potential genetic vulnerability of children with ASD to oxidative stress [11,54-56], environmental factors contributing to it [11,57,58], as well as deficient mechanisms of DNA repair [59].

Excess ROS in an organism oxidizes biomolecules, such as lipids,yielding malondialdehyde (MDA) and F2-isoprostanes; the oxidation ofDNA results in 8-hydroxy-2’-deoxyguanosine (8-OHdG), and o,o’-dityrosine (diY) is a product of protein oxidation. The products of oxidation by ROS are excreted in measurable concentrations in urine,which can be determined as biomarkers of oxidative stress.

[45] Dora Il'yasova, Peter Scarbrough, Ivan Spasojevic.Urinary biomarkers of oxidative status.ClinicaChimicaActa 413 (2012) 1446–1453 doi:10.1016/j.cca.2012.06.012

[28] Thomas J. van 'tErve, Maria B. Kadiiska, Stephanie J. London, Ronald P. Mason. Classifying oxidative stress by F2-isoprostane levels across human diseases:A meta-analysis. Redox Biology 12 (2017) 582–599

Although blood has been used in the analysis of oxidative stress biomarkers (OSBs), urine is a preferred matrix due to the non-invasiveness of sampling and those OSBs excrete primarily in urine. Due to the fact that variability in urinary concentrations of OSBs in different studies is less well known…

The results of different studies suggest that creatinine normalization decreased intra- and inter-individual variabilities in urinary OSB concentrations

Reviewer 3 Report

The mauscript present the analysis of 4 urinary biomarkers of autism spectrum disorders (ASD). The approach is useful for ASD however many other biomarkers in urine related to oxydative stress have been already investigated. The sizes of the experimental groups are very different, and  the number of subjects in the control group shoudld be increased in order to reach a size similar to that of ASD group. The measurement, analysis of urine sample, is not invasive and it can easily be done otherwise the risk of bias towards significant results might be masked for such differences in the sample sizes of the two experimental groups or at least it should be minimized.

ASD as a clear male to female predominance and it's crucial to maintain a similar sex distribution in the control group, however in the present study, male participants were the 89% in the ASD group and only 48% in the control group. This could also represent a source of bias and should be minimized with the proper sampling procedure to avoid bias.

The support vector machines (SVM) with the three different kernel functions (linear, radial, polynomial) should be described in details, no mention of it in the Methods section, making the reproducibility of the results very difficult.

The reason to select the four urinary biomarkers of oxidative stress shoudl be justified and supported by scientific evidence.

Published data of oxydative stress markers in urine samples from ASD patients should be cited:

1: Liu A, Zhou W, Qu L, He F, Wang H, Wang Y, Cai C, Li X, Zhou W, Wang M. Altered Urinary Amino Acids in Children With Autism Spectrum Disorders. Front Cell Neurosci. 2019 Jan 25;13:7. doi: 10.3389/fncel.2019.00007. eCollection 2019. PubMed PMID: 30733669; PubMed Central PMCID: PMC635412.

2: Li C, Shen K, Chu L, Liu P, Song Y, Kang X. Decreased levels of urinary free amino acids in children with autism spectrum disorder. J Clin Neurosci. 2018 Aug;54:45-49. doi: 10.1016/j.jocn.2018.05.001. Epub 2018 May 28. PubMed PMID: 29853226.

3: Anwar A, Abruzzo PM, Pasha S, Rajpoot K, Bolotta A, Ghezzo A, Marini M, Posar A, Visconti P, Thornalley PJ, Rabbani N. Advanced glycation endproducts, dityrosine and arginine transporter dysfunction in autism - a source of biomarkers for clinical diagnosis. Mol Autism. 2018 Feb 19;9:3. doi:10.1186/s13229-017-0183-3. eCollection 2018. PubMed PMID: 29479405; PubMed Central PMCID: PMC5817812. 

4: Fuentes Albero M, Cauli O. Homocysteine Levels in Autism Spectrum Disorder: A Clinical Update. Endocr Metab Immune Disord Drug Targets. 2018;18(4):289-296. doi: 10.2174/1871530318666180213110815. Review. PubMed PMID: 29437021.

5: Lussu M, Noto A, Masili A, Rinaldi AC, Dessì A, De Angelis M, De Giacomo A, Fanos V, Atzori L, Francavilla R. The urinary (1) H NMR metabolomics profile of an italian autistic children population and their unaffected siblings. Autism Res. 2017 Jun;10(6):1058-1066. doi: 10.1002/aur.1748. Epub 2017 Mar 11. PubMed PMID: 28296209.

6: Yui K, Tanuma N, Yamada H, Kawasaki Y. Decreased total antioxidant capacity has a larger effect size than increased oxidant levels in urine in individuals with autism spectrum disorder. Environ Sci Pollut Res Int. 2017 Apr;24(10):9635-9644. doi: 10.1007/s11356-017-8595-3. Epub 2017 Mar 1. PubMed PMID: 28247276.

The descriptive statistics of the  Autism Diagnostic Observation Schedule (ADOS) and its subdomains should be represented: mean, standard deviation, range, etc

.The criteria to allocate the ASD participants into mild, moderate,sever deficits categories of social communication and behavioral flexibility should be described in details as well as the criteria to repetitive/stereotyped behaviours, which is one of the core symtoms of ASD. The criteria for ASD diagnosis should be specified as well.

Author Response

Reviewer3

The mauscript present the analysis of 4 urinary biomarkers of autism spectrum disorders (ASD). The approach is useful for ASD however many other biomarkers in urine related to oxydative stress have been already investigated. The sizes of the experimental groups are very different, and  the number of subjects in the control group shoudld be increased in order to reach a size similar to that of ASD group. The measurement, analysis of urine sample, is not invasive and it can easily be done otherwise the risk of bias towards significant results might be masked for such differences in the sample sizes of the two experimental groups or at least it should be minimized.

In the response, the reviewer highlights some substantially meaningful comments, while in other places, in our opinion, overlook the already written one. It is overlooked, for example, that we have explained in the introduction why we chose to select biomarkers and justify them. It is also overlooked that some articles are required to quote them, have already been quoted (for example, "Advanced glycation end products, dityrosine and arginine transporter dysfunction and autism - a source of biomarkers for clinical diagnosis").

In the first version of our manuscript we had a control group of 21 subjects. We already undertook efforts to obtain additional data for the control group, which is now 47, which allowed us to generate two separate sets – a training dataset and a validation dataset (untouched before validation testing). As could be expected we found that the prediction accuracy in the validation sample was significantly lower. It was however still above chance, indicating that the chosen biomarkers can contribute to a better understanding of the role of oxidative stress in ASD.

ASD as a clear male to female predominance and it's crucial to maintain a similar sex distribution in the control group, however in the present study, male participants were the 89% in the ASD group and only 48% in the control group. This could also represent a source of bias and should be minimized with the proper sampling procedure to avoid bias.

We are aware of this discrepancy but the markers which we determined in our study were not significantly affected by gender, so in this sense it is not so important to match gender strictly.The concentrations and profiles of oxidative stress markers are not significantly affected by gender, age, or BMI.

This statement is supported by the results of the following studies:

[70] Maria-Pilar Martinez-Moral, Kurunthachalam KannanHow stable is oxidative stress level? An observational study of intra- and inter-individual variability in urinary oxidative stress biomarkers of DNA, proteins, and lipids in healthy individuals. Environment International 123 (2019) 382–389.

[69] M. ElisabettaZanolin, Paolo Girardi, Paolo Degan, Marta Rava, Mario Olivieri, Gianfranco Di Gennaro, MorenaNicolis, Roberto De Marco.Measurement of a urinary marker (8-hydroxydeoxyguanosine, 8-OHdG) of DNA oxidative stress in epidemiological surveys: a pilot study.Int J Biol Markers 2015; 30(3): e341-e345

The support vector machines (SVM) with the three different kernel functions (linear, radial, polynomial) should be described in details, no mention of it in the Methods section, making the reproducibility of the results very difficult.

An explanation of the kernel function for SVMs - the simplest way is by directing the reviewer to existing literature - we apply two articles for this clarification:

 - Ben-Hur A, Ong CS, Sonnenburg S, Scholkopf B, Ratsch G (2008) Support Vector Machines andKernels for Computational Biology. PLoSComputBiol 4(10): e1000173. doi:10.1371/journal.pcbi.1000173and

- Paulo Gaspar, Jaime Carbonelland Jose Luıs Oliveira.On the parameter optimization of Support Vector Machines for binary classification.Journal of Integrative Bioinformatics, 9(3):201, 2012).

The reviewer’s comment on the ability to repeat our study is not justified, as we clearly state the script in the R + programming language, when it is easy to run on new data.

The reason to select the four urinary biomarkers of oxidative stress shoudl be justified and supported by scientific evidence.

Oxidative damage produced by reactive oxygen species (ROS) has been implicated in the etiology and pathologyof many health conditions, including a large number of chronic diseases. Urinary biomarkers of oxidative statuspresent a great opportunity to study redox balance in human populations. With urinary biomarkers, specimencollection is non-invasive and the organic/metal content is low, which minimizes the artifactual formation ofoxidative damage to molecules in specimens. Also, urinary levels of the biomarkerspresent intergraded indices of redox balance over a longer period oftime compared to blood levels, which may make them more sensitiveto predicting chronic conditions while also decreasing intra-individualvariability of the measurements.

Urinary F2-isoprostanes: this is a well-studied and validated biomarker of oxidative status, with known inter- and intra-individualvariations. However, there is some concern that inter-individualvariation in F2-isoprostane hydrolysis form esterified lipids thatcould contribute to the differences in their steady-state urinarylevels

Urinary 8-oxodG: 8-oxodG has been validated as a biomarker foroxidative stress in animal models, but so far this has not beenshown in human studies. The impact of inter-individual variabilityin DNA repair capacity on the levels of 8-oxodG in urine remainsunknown. Intra-individual variability is also uncertain. However,this biomarker can be useful as a monitoring tool.

Urinary dityrosine: Promising understudied biomarker, such as dityrosine, which reflect oxidative protein damage.Addition of this biomarker to the panel of validated indices of oxidative status diversify the existing tools in studying the role ofoxidative status in human health and disease.

Urinary HEL: N-epsilon-hexanoyl-lysine (HEL) is a novel lipid peroxidation biomarker which isderived from the oxidation of omega-6 unsaturated fatty acid.

Scientific literature:

Thomas J. van 'tErve, Maria B. Kadiiska, Stephanie J. London, Ronald P. Mason. Classifying oxidative stress by F2-isoprostane levels across human diseases:A meta-analysis. Redox Biology 12 (2017) 582–599]

Dora Il'yasova, Peter Scarbrough, Ivan Spasojevic. Urinary biomarkers of oxidative status.ClinicaChimicaActa 413 (2012) 1446–1453 doi:10.1016/j.cca.2012.06.012

Isabella Dalle-Donne, Ranieri Rossi, Roberto Colombo, Daniela Giustarini, and Aldo Milzani. Biomarkers of Oxidative Damage in Human Diseases .Clinical Chemistry (2006)52;4:601–623 DOI: 10.1373/clinchem.2005.061408

Kazuo Sakai, Satoko Kino, Aino Masuda, Masao Takeuchi, TairinOchi,JoskoOsredkar, Barbara Rejc, KsenijaGersak,NarasimhanRamarathnam, and Yoji Kato. Determination of HEL (Hexanoyl-LysineAdduct): A Novel Biomarker for Omega-6PUFA Oxidation. Y. Kato (ed.),LipidHydroperoxide-Derived Modification of Biomolecules,Subcellular Biochemistry 77, DOI 10.1007/978-94-007-7920-4_5,©Springer Science+Business Media Dordrecht 2014

Published data of oxydative stress markers in urine samples from ASD patients should be cited

In the articles that follow, other parameters are presented, but in total, they have also presented results in these studies by comparing the disparity between groups according to gender, and we quote these articles when we refer to the weaknesses of our study.

[71] 1: Liu A, Zhou W, Qu L, He F, Wang H, Wang Y, Cai C, Li X, Zhou W, Wang M. Altered Urinary Amino Acids in Children With Autism Spectrum Disorders. Front Cell Neurosci. 2019 Jan 25;13:7. doi: 10.3389/fncel.2019.00007. eCollection 2019. PubMed PMID: 30733669; PubMed Central PMCID: PMC6354128.

In this article, the authors present the results of a study of free amino acids in urine of ASD children. We read this article and we see that the study was conducted in two phases. From the session Participants we can see that the discovery cohort included 28 ASD children (26 males = 92.8%) and TD children (19 males = 73.2%); the validation cohort consisted of 29 ASD children (27 males = 93.1%) and 41 TD children (7 males = 29.2%).

2: Li C, Shen K, Chu L, Liu P, Song Y, Kang X. Decreased levels of urinary free amino acids in children with autism spectrum disorder. J Clin Neurosci. 2018 Aug;54:45-49. doi: 10.1016/j.jocn.2018.05.001. Epub 2018 May 28. PubMed PMID: 29853226.

In this article, the authors present the results of a study of amino acids in urine as in previous article. Our study does not address these parameters, and therefore we did not cite this study.

3: Anwar A, Abruzzo PM, Pasha S, Rajpoot K, Bolotta A, Ghezzo A, Marini M, Posar  A, Visconti P, Thornalley PJ, Rabbani N. Advanced glycation endproducts, dityrosine and arginine transporter dysfunction in autism – a source of biomarkers for clinical diagnosis. Mol Autism. 2018 Feb 19;9:3. doi: 10.1186/s13229-017-0183-3. eCollection 2018. PubMed PMID: 29479405; PubMed Central PMCID: PMC5817812.

Alreadycitatedin previoustextas ref.34 (nowchanged37)

4: FuentesAlbero M, Cauli O. Homocysteine Levels in Autism Spectrum Disorder: A  Clinical Update. Endocr Metab Immune Disord Drug Targets. 2018;18(4):289-296. doi: 10.2174/1871530318666180213110815. Review. PubMed PMID: 29437021.

In our study, we did not have homocysteine in the protocol, so we do not know why the reviewer pointed out the requirement to quote this article. We know from the literature that it is an interesting parameter and we will include it in our research in future.

[72] 5: Lussu M, Noto A, Masili A, Rinaldi AC, Dessì A, De Angelis M, De Giacomo A, Fanos V, Atzori L, Francavilla R. The urinary (1) H-NMR metabolomics profile of an italian autistic children population and their unaffected siblings. Autism Res. 2017 Jun;10(6):1058-1066. doi: 10.1002/aur.1748. Epub 2017 Mar 11. PubMed PMID: 28296209.

This article describes a study using nuclear magnetic resonance-based metabolomic analysis of the urine. In terms of the results of analyzes, it is very interesting and it opens up a lot of research opportunities. We quote this study in the part where we list the studies with disproportionate gender representation.

[73] 6: Yui K, Tanuma N, Yamada H, Kawasaki Y. Decreased total antioxidant capacity has a larger effect size than increased oxidant levels in urine in individuals with autism spectrum disorder. Environ Sci Pollut Res Int. 2017 Apr;24(10):9635-9644. doi: 10.1007/s11356-017-8595-3. Epub 2017 Mar 1. PubMed PMID: 28247276.

Following the recommendation of the reviewer, we have read the article of these authors and put it in the discussion, and also quote them.In this case too, there is a gender disparity when it comes to the ASD group and a group of healthy children.

The descriptive statistics of the Autism Diagnostic Observation Schedule (ADOS) and its sub-domains should be represented: mean, standard deviation, range, etc

Dealing with children of different ages and using different modules (so their results are not directly comparable), ADOS was performed, but the results of the test were not shown because patients were grouped according to the criteria detailed below and summarized by DSM-5. On this basis, patients were classified into three groups according to the degree of disability.

Given the degree of deficit, obstacles disorders can be divided into children with autism disorders with a lighter, moderate or severe deficit in social communication and social interaction.

1)      Easier deficit in social communication and social interaction. A child has difficulty in making contact. Inadequately responds to other people's social initiatives and may indicate a reduced interest in interacting with others. Inadequately establishing social relationships, there is a lack of reciprocity in communication, poor integrationverbal and non-verbal communication, and difficulties in adapting behavior to different social circumstances.

2)    Moderate Deficit in Social Communication and Social Interaction The child has significant shortcomings in verbal and non-verbal social communication. It is noticeable that the establishment of social relations is limited, inadequate response to the social initiatives of others. The child expresses the disability and shares the interests and emotions with others, inadequately establishes an eye contact. His body language is unusual, shows deficits in understanding and using gestures, and difficulties in participating in a symbolic game.

3)    The worse deficit in social communication and social interaction The child has very significant deficits in the area of verbal and non-verbal social communication, causing significant disruption to its functioning. It is very limited in establishing social relations and responding to the social initiatives of others to a minimum. He cannot start or respond to the initiative after social interaction, facial mimics and non-verbal communication are absent, they are not interested in peers.

A) Lighter deficit in the field of behavior, interests and activities. The child's inflexible / rigid behavior causes deviations in the functioning in one or more areas. A child has difficulty in moving between activities. Organizational and planning issues hinder child's independence.

B) Moderate deficit in the field of behavior, interests and activity The child's inflexible / rigid behavior causes significant deviations in adapting to changes. The child is over-occupied with interests and / or behaviors that occur so often, that they are also obvious to a random observer and influence the child's activity in several different areas. A child shows hardship if he has to change his or her behavior. Adapt to environmental requirements

 C) A more severe deficit in the field of behavior, interests and activities.The inflexible / rigid behavior of a child causes very important deviations in adapting to change. The child is such a preoccupied interest and / or behavior that it prevents thisits operation in several different areas. The child shows a lot of hardship if he has to change his / adapt to environmental requirements.

The criteria to allocate the ASD participants into mild, moderate,sever deficits categories of social communication and behavioral flexibility should be described in details as well as the criteria to repetitive/stereotyped behaviours, which is one of the core symtoms of ASD. The criteria for ASD diagnosis should be specified as well.

For ASD, the diagnostic criteria from ICD-10-CM [It is a manual that reflects current state of knowledge and consensus among leaders in the field] are followed to assist in diagnostic decision making. Specific recording procedures were used with the diagnostic criteria to provide guidance in selecting the most appropriate code. The revised diagnosis represents a new,more accurate, and medically and scientifically useful way of diagnosing individuals with autism-related disorders.

Severity levels for autism spectrum disorder

Severitylevel

Social communication

Restricted,    repetitivebehaviors

Level 3
  "Requiringverysubstantialsupport”

Level 2
  "Requiringsubstantialsupport”

Level 1
  "Requiringsupport”

Severe deficits in verbalandnonverbal social communicationskillscause   severe impairments in functioning, verylimitedinitiationof social   interactions, andminimalresponse to social overturesfromothers. Forexample, a   personwithfewwordsofintelligiblespeechwhorarelyinitiatesinteractionand,   whenhe or shedoes, makesunusualapproaches to meetneedsonlyandresponds to   onlyverydirect social approaches

Inflexibilityofbehavior, extremedifficultycopingwithchange, or   otherrestricted/repetitivebehaviorsmarkedlyinterferewithfunctioning in   allspheres. Greatdistress/difficultychangingfocus or action.

Markeddeficits in verbalandnonverbal social communicationskills; social   impairmentsapparentevenwithsupports in place; limitedinitiationof social   interactions; andreduced or  abnormalresponses to social   overturesfromothers. Forexample, a personwhospeakssimplesentences,   whoseinteraction is limited  to narrowspecialinterests,   andwhohasmarkedly odd nonverbalcommunication.

Inflexibilityofbehavior, difficultycopingwithchange, or   otherrestricted/repetitivebehaviorsappearfrequentlyenough to beobvious to   thecasualobserverandinterferewithfunctioning in a varietyofcontexts.   Distressand/or difficultychangingfocus or action.

Withoutsupports in place, deficits in social   communicationcausenoticeableimpairments. Difficultyinitiating social   interactions, andclearexamplesofatypical or unsuccessfulresponse to social   overturesofothers. Mayappear to havedecreasedinterest in social interactions.   Forexample, a personwho is able to speak in fullsentencesandengages in   communicationbutwhose to- and-froconversationwithothersfails,   andwhoseattempts to make friends are odd andtypicallyunsuccessful.

Inflexibilityofbehaviorcausessignificant interference withfunctioning in   one or more contexts. Difficultyswitchingbetweenactivities.   Problemsoforganizationandplanninghamperindependence

People with ASD tend to have communication deficits, such as responding inappropriately in conversations, misreading nonverbal interactions, or having difficulty building friendships appropriate to theirage. In addition, people with ASD may be overly dependent on routines, highly sensitive to changes intheir environment, or intensely focused on inappropriate items. Again, the symptoms of people withASD will fall on a continuum, with some individuals showing mild symptoms and others having muchmore severe symptoms.
More recently, the largest and most up-to-date study, published by Huerta, et al, in the October 2012issue of American Journal of Psychiatry, provided the most comprehensive assessment of the DSM-5criteria for ASD based on symptom extraction from previously collected data. The study found thatDSM-5 criteria identified 91 percent of children with clinical DSM-IV PDD diagnoses, suggesting that most children with DSM-IV PDD diagnoses will retain their diagnosis of ASD using the new criteria. Several other studies, using various methodologies, have been inconsistent in their findings.

Round  2

Reviewer 1 Report

No any comment.

Reviewer 2 Report

Authors made significative improvements according to my comments.

Reviewer 3 Report

Agree to review